# Coagulation factors directly cleave SARS-CoV-2 spike and enhance viral entry

Edward R Kastenhuber[1], Marisa Mercadante[1], Benjamin Nilsson-Payant[2], Jared L Johnson[1], Javier A Jaimes[3], Frauke Muecksch[4], Yiska Weisblum[4], Yaron Bram[5], Vasuretha Chandar[5], Gary R Whittaker[3], Benjamin R tenOever[2], Robert E Schwartz[5,6], Lewis Cantley[1]*

[1]Meyer Cancer Center, Department of Medicine, Weill Cornell Medical College, New York, United States; [2]Department of Microbiology, New York University - Langone Health, New York, United States; [3]Department of Microbiology and Immunology, Cornell University, Ithaca, United States; [4]Laboratory of Retrovirology, The Rockefeller University, New York, United States; [5]Division of Gastroenterology and Hepatology, Department of Medicine, Weill Cornell Medicine, New York, United States; [6]Department of Physiology, Biophysics and Systems Biology, Weill Cornell Medicine, New York, United States

**Abstract** Coagulopathy is a significant aspect of morbidity in COVID-19 patients. The clotting cascade is propagated by a series of proteases, including factor Xa and thrombin. While certain host proteases, including TMPRSS2 and furin, are known to be important for cleavage activation of SARS-CoV-2 spike to promote viral entry in the respiratory tract, other proteases may also contribute. Using biochemical and cell-based assays, we demonstrate that factor Xa and thrombin can also directly cleave SARS-CoV-2 spike, enhancing infection at the stage of viral entry. Coagulation factors increased SARS-CoV-2 infection in human lung organoids. A drug-repurposing screen identified a subset of protease inhibitors that promiscuously inhibited spike cleavage by both transmembrane serine proteases and coagulation factors. The mechanism of the protease inhibitors nafamostat and camostat may extend beyond inhibition of TMPRSS2 to coagulation-induced spike cleavage. Anticoagulation is critical in the management of COVID-19, and early intervention could provide collateral benefit by suppressing SARS-CoV-2 viral entry. We propose a model of positive feedback whereby infection-induced hypercoagulation exacerbates SARS-CoV-2 infectivity.

*For correspondence:
lcantley@med.cornell.edu

## Editor's evaluation

This study examines the potential role host proteases involved in coagulation may play in proteolytic processing of the SARS CoV-2 virus spike protein, which is required for viral entry. Serine protease inhibitors such as nafamostat and camostat may limit viral entry into host cells and also be useful to treat coagulopathy in patients with SARS CoV-2 infection, particularly if treatment is initiated early.

## Introduction

SARS-CoV-2 emerged into the human population in late 2019 and has evolved into a devastating global health crisis. Despite the recent success of vaccines (*Baden et al., 2021*; *Polack et al., 2020*), the limited world-wide vaccine distribution (*Kwok et al., 2021*; *Lin et al., 2020*; *Nhamo et al., 2021*; *So and Woo, 2020*), the emergence of viral variants (*Wang et al., 2021*; *Weisblum et al., 2020*), and the repeated SARS-like zoonotic outbreaks over the last 20 years (*Cheng et al., 2007*; *Ge et al.,*

*2013*; *Menachery et al., 2015*) underscore the urgent need to develop antivirals for coronavirus (*Pan et al., 2021*).

In addition to attachment to specific receptors on target cells, coronaviruses require proteolytic processing of the spike protein by host cell proteases to facilitate membrane fusion and viral entry (*Glowacka et al., 2011*; *Jaimes et al., 2020c*; *Walls et al., 2020*). In SARS-CoV-2, host cell proteases act on two sites residing at the S1/S2 subunit boundary and at the S2' region proximal to the fusion peptide (*Belouzard et al., 2009*; *Hoffmann et al., 2020a*; *Jaimes et al., 2020b*; *Millet and Whittaker, 2014*). S1/S2 site cleavage opens up the spike trimer and exposes the S2' site, which must be cleaved to allow for the release of the conserved fusion peptide (*Benton et al., 2020*; *Madu et al., 2009*). While the prevailing model suggests that furin cleaves the S1/S2 site and TMPRSS2 cleaves the S2' site (*Bestle et al., 2020*), it remains unclear to what extent other proteases may be involved (*Hoffmann et al., 2021*; *Ou et al., 2020*).

TMPRSS2 is an important host cell factor in proteolytic activation across multiple coronaviruses (*Hoffmann et al., 2020a*; *Jaimes et al., 2019*). TMPRSS2 knockout or inhibition reduces infection in mouse models of SARS and MERS (*Iwata-Yoshikawa et al., 2019*; *Zhou et al., 2015*). More recently, TMPRSS2 has been highlighted as a drug target for SARS-CoV-2 (*Hoffmann et al., 2020a*; *Hoffmann et al., 2020b*).

Furin activity is not essential to produce infectious particles (*Tang et al., 2021a*) and furin is not necessary for cell fusion (*Papa et al., 2021*), but deletion of the S1/S2 site attenuates SARS-CoV-2 in vivo (*Johnson et al., 2021*). Proteolytic activation of envelope proteins presumably coordinates target cell engagement and envelope conformational changes leading to fusion. Furin cleavage during viral biogenesis, before release of viral particles, may render SARS-CoV-2 spike less stable in solution and reduce the likelihood to reach and interact with target cells (*Amanat et al., 2021*; *Berger and Schaffitzel, 2020*; *Wrobel et al., 2020*). Although the S1/S2 site is often referred to as the 'furin site' (*Johnson et al., 2021*), the full spectrum of proteases that catalyze biologically relevant activity in the lung remains incompletely defined.

Proteases also orchestrate the coagulation pathway, via a series of zymogens that are each activated by a chain reaction of proteolytic processing. Coagulopathy and thromboembolic events have emerged as a key component of COVID-19 pathogenesis (*McGonagle et al., 2020*). Comorbidities associated with severe COVID-19 are also linked to dysregulated blood clotting (*Zhou et al., 2020*). Patients with a history of stroke prior to infection have nearly twice the risk of in-hospital mortality (*Qin et al., 2020*). Upon hospital admission, elevated D-dimer levels (an indicator of fibrinolysis and coagulopathy) and low platelet counts (an indicator of consumptive coagulopathy) are predictive biomarkers of severe disease and lethality in COVID-19 patients (*Zhou et al., 2020*). Systemic activity of clotting factors V, VIII, and X are elevated in severe COVID-19 disease (*Stefely et al., 2020*). While early phase disease is typically restricted to a local pulmonary hypercoagulable state, late-stage disease may be accompanied by systemic DIC, stroke, and cardio-embolism (*Huang et al., 2020*; *Kipshidze et al., 2020*; *McGonagle et al., 2020*; *Tsivgoulis et al., 2020*). Ischemic stroke occurred in approximately 1% of hospitalized COVID-19 patients, and strikingly, a fraction of them experienced stroke even prior to onset of respiratory symptoms (*Yaghi et al., 2020*).

In a drug-repurposing effort to target TMPRSS2, we observed that multiple direct-acting anticoagulants have anti-TMPRSS2 off-target effects. We subsequently investigated overlap in substrate specificity between TMPRSS2, factor Xa, and thrombin. Circulating proteases involved in blood clotting can cleave and activate SARS-CoV-2 spike, enhancing infection, specifically at the stage of viral entry. We propose that the serine protease inhibitor nafamostat may incorporate a combined mechanism in the treatment of COVID-19 through inhibition of TMPRSS2 and coagulation factors.

## Results

### Serine protease inhibitors suppress SARS-CoV-2 entry via inhibition of TMPRSS2

We developed a fluorescence resonance energy transfer (FRET)-based protease enzymatic assay based on peptides containing either the S1/S2 or S2' cleavage sites of SARS-CoV-2 spike (*Figure 1A*, *Figure 1—figure supplement 1*). Upon cleavage, the liberated 5-FAM emits fluorescent signal proportional to the quantity of product (*Figure 1—figure supplement 1A*). Camostat and nafamostat

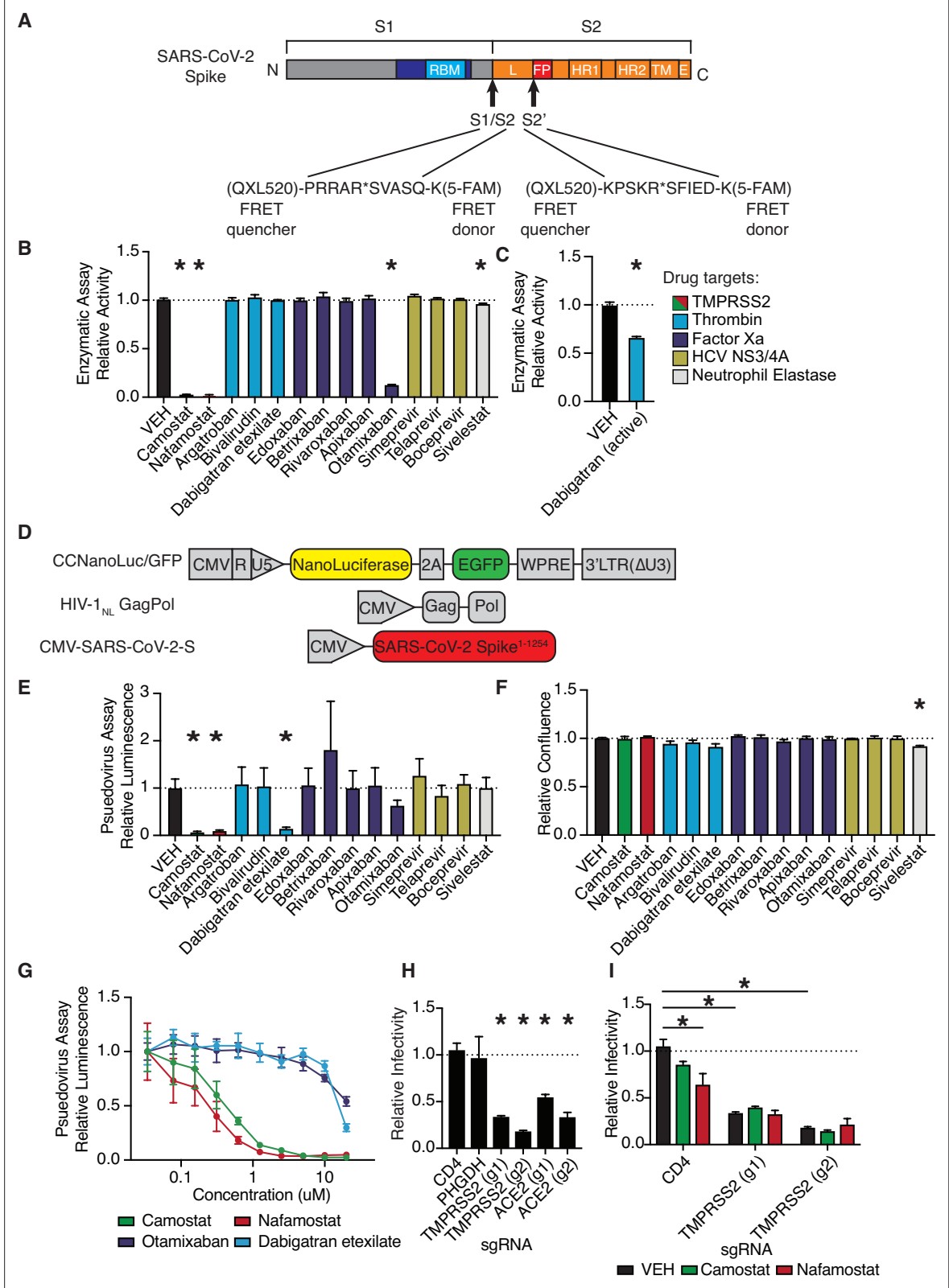

**Figure 1.** Anticoagulant serine protease inhibitors suppress SARS-CoV-2 entry via inhibition of TMPRSS2. (**A**) Peptides derived from two known cleavage sites of SARS-CoV-2 spike were designed with C-terminal fluorophore 5-FAM and N-terminal fluorescence resonance energy transfer (FRET) quencher QXL-520. (**B**) FDA-approved and investigational serine protease inhibitors were screened by enzymatic assay to inhibit TMPRSS2 cleavage of SARS-CoV-2 S1/S2 peptide substrate. Relative change in fluorescence with respect to DMSO vehicle is shown. Colors indicate the described target

*Figure 1 continued on next page*

*Figure 1 continued*

of the drugs screened. All drugs screened at 10 μM final concentration. (**C**) Active form of dabigatran in enzymatic assay for TMPRSS2 inhibition. Relative fluorescence with respect to its corresponding 0.1 N HCl vehicle is shown. (**D**) Schematic of constructs used to generate SARS-CoV-2 spike-pseudotyped/HIV-1-based particles. (**E**) Calu3 cells were treated with 10 μM of the indicated drugs for 24 hr prior to infection with HIV-1$_{NL}$/SARS-CoV-2 pseudovirus. Media was changed at 24 hr post infection and pseudoviral entry was measured by nanoluciferase luminescent signal at 40 hr. (**F**) Calu3 cells treated with 10 μM of the indicated drugs were monitored for confluence by Incucyte for 40 hr. (**G**) Pseudoviral entry was measured by nanoluciferase luminescent signal in Calu3 cells treated various concentrations of the indicated drugs for 4 hr prior to infection with SARS-CoV-2 pseudovirus. (**H**) Caco2 cells were infected with lenti-Cas9-blast and U6-sgRNA-EFS-puro-P2A-tRFP and selected. Neutral controls targeting CD4 (not endogenously expressed) or PHGDH intron 1, two sgRNAs each targeting different regions of ACE2 and TMPRSS2 were included. Cells were subsequently infected with HIV-1$_{NL}$/SARS-CoV-2 pseudovirus. (**I**) Caco2 cells co-expressing Cas9 and sgRNAs targeting CD4 (not expressed) or TMPRSS2 were treated with 10 μM camostat, nafamostat, or DMSO vehicle. N = 3, *p < 0.05, two-tailed t-test. Data represented as mean ± SEM.

The online version of this article includes the following source data and figure supplement(s) for figure 1:

**Source data 1.** Data and summary statistics for enzymatic and pseudovirus assays.

**Figure supplement 1.** Optimization of fluorescence resonance energy transfer (FRET) enzymatic assay.

**Figure supplement 2.** Further characterization of HIV-1/SARS-CoV-2 pseudovirus.

**Figure supplement 3.** Further characterization of rVSVΔG/SARS-CoV-2 pseudovirus.

**Figure supplement 4.** Evidence of CRISPR knockout efficiency.

resulted in strong inhibition of TMPRSS2 (*Figure 1B*), as expected (*Hoffmann et al., 2020a*; *Hoffmann et al., 2020b*). We also identified that otamixaban and the active form of dabigatran (but not its prodrug dabigatran etexilate) inhibit TMPRSS2 enzymatic activity in vitro (*Figure 1B–C*).

To explore these candidates in a cell-based functional assay of spike protein, SARS-CoV-2 S-pseudotyped HIV-1 particles were employed to infect human lung Calu3 cells (*Figure 1D*; *Schmidt et al., 2020*). Consistent with the TMPRSS2 enzymatic assay, camostat, nafamostat, otamixaban, and dabigatran etexilate suppressed pseudoviral entry, as indicated by nanoluciferase luminescent signal (*Figure 1E*). No effects on relative cell growth were observed at the same timepoint in Calu3 (*Figure 1F*) or A549 cells (data not shown), confirming that reduced luminescent signal was not due to cytotoxicity. A dose-response experiment with select protease inhibitors revealed a submicromolar IC50 for camostat and nafamostat and IC50s in the 10–20 μM range for otamixaban and dabigatran in Calu3 cells (*Figure 1G*).

Using A549 cells with or without ectopic ACE2 expression, we confirmed that HIV-1$_{NL}$/SARS-CoV-2 pseudovirus infection is dependent on ACE2, while infection with HIV-1$_{NL}$ pseudotyped instead with VSV G envelope protein is not ACE2 dependent (*Figure 1—figure supplement 2*). Caco2 cells, which endogenously express ACE2 and TMPRSS2, show greater susceptibility to SARS-CoV-2 S-pseudotyped HIV-1$_{NL}$, but equivalent susceptibility to VSV G-pseudotyped HIV-1$_{NL}$, when compared to A549/ACE2 cells (*Figure 1—figure supplement 2*).

To further validate these results in an alternative pseudovirus system, we used recombinant G protein-deficient vesicular stomatitis virus (rVSVΔG) pseudotyped with SARS-CoV-2-S (*Figure 1—figure supplement 3A*), yielding pseudovirus dependent on spike for cell entry (*Figure 1—figure supplement 3B*). The antiviral effects of the four candidate protease inhibitors were confirmed in the VSV pseudovirus system in multiple cell lines, and response was associated with TMPRSS2 expression (*Figure 1—figure supplement 3C-F*).

We aimed to determine whether the effects of camostat and nafamostat are indeed TMPRSS2-dependent, or if other unidentified cellular proteases can compensate for TMPRSS2 suppression. To do so, we knocked out TMPRSS2 in ACE2$^+$ TMPRSS2$^+$ Caco2 cells and found that susceptibility to pseudovirus was significantly reduced, comparable to knockout of ACE2 (*Figure 1H*, *Figure 1—figure supplement 4*). Furthermore, both camostat and nafamostat reduce pseudovirus entry into control Caco2 cells harboring control sgRNA, but this trend was lost in cells with two independent TMPRSS2-targeting sgRNAs (*Figure 1I*). These data indicate that, in the absence of exogenous proteases, TMPRSS2 is a critical host enzyme activating SARS-CoV-2 spike in TMPRSS2$^+$ cells and that TMPRSS2 is the primary target of camostat and nafamostat in these conditions.

## Coagulation factors directly cleave SARS-CoV-2 spike

Anticoagulants are highly represented among FDA-approved drugs that target proteases, and among the hits from the screen described above. The off-target effects of anticoagulants on TMPRSS2 imply

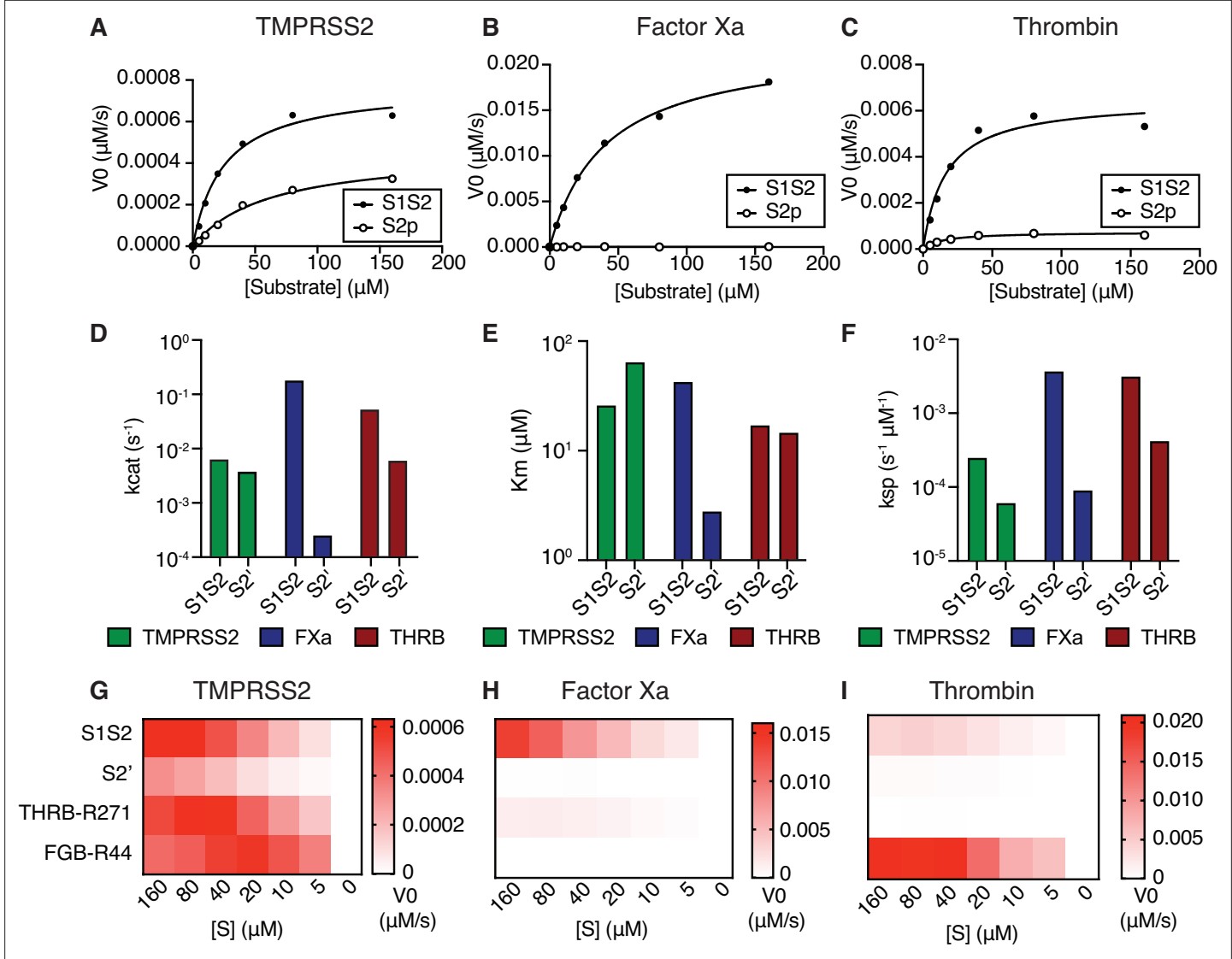

**Figure 2.** Coagulation factors directly cleave SARS-CoV-2 spike. Initial velocities for the cleavage of SARS-CoV-2 spike S1/S2 and S2' peptide substrates by (**A**) TMPRSS2, (**B**) factor Xa, and (**C**) thrombin were measured over a range of 0–160 μM substrate. From initial velocity values, enzyme kinetic constants (**D**) turnover rate $K_{cat}$ ($s^{-1}$), (**E**) affinity constant $K_m$, and (**F**) specificity constant ($K_{cat}/K_m$) were obtained for the indicated enzymes with S1/S2 and S2' peptides. (**G–I**) Heatmaps depict the initial velocity $V_0$ of cleavage of the indicated peptide substrates and concentrations by (**G**) TMPRSS2, (**H**) factor Xa, and (**I**) thrombin.

The online version of this article includes the following source data and figure supplement(s) for figure 2:

**Source data 1.** Data and summary statistics for enzymatic assays.

**Figure supplement 1.** Fluorescence resonance energy transfer (FRET) enzymatic assay with modified peptide substrates.

that these small molecules can interact with the active sites of TMPRSS2 in a similar manner to coagulation factors. This led us to hypothesize that coagulation factors may interact with some of the same substrates as TMPRSS2, including SARS-CoV-2 spike.

To determine the properties of enzyme-substrate relationships, TMPRSS2, factor Xa, and thrombin cleavage of S1/S2 and S2' peptides were determined over a range of 0–160 μM peptide substrate (*Figure 2A–C*, *Table 1*). Surprisingly, factor Xa catalyzed S1/S2 cleavage more than an order of magnitude faster than TMPRSS2 (*Figure 2A–B and D*), although factor Xa showed lower affinity (higher $K_m$) compared with TMPRSS2 to the S1/S2 peptide (*Figure 2A–B and E*). Thrombin has greater affinity (lower $K_m$) than TMPRSS2 and factor Xa for the S1/S2 substrate (*Figure 2E*) and performs S1/

**Table 1.** Kinetics of SARS-CoV-2 spike peptide substrate cleavage.

Kinetic constants obtained from initial velocity studies with varying concentrations of SARS-CoV-2 spike S1/S2 and S2′ peptide substrates. Each estimate is based on seven different concentrations of substrate in 1:2 serial dilution (0–160 µM).

| Enzyme | Substrate | $V_{max}$ (µM/s) | $K_{cat}$ (s$^{-1}$) | $K_m$ (µM) | $K_{sp}$ (s$^{-1}$ µM$^{-1}$) |
|---|---|---|---|---|---|
| TMPRSS2 | S1/S2 | 7.71E-04 | 6.17E-03 | 24.71 | 2.50E-04 |
| TMPRSS2 | S2′ | 4.60E-04 | 3.68E-03 | 60.94 | 6.04E-05 |
| Factor Xa | S1/S2 | 2.24E-02 | 1.79E-01 | 40.35 | 4.43E-03 |
| Factor Xa | S2′ | 3.04E-05 | 2.43E-04 | 2.711 | 8.97E-05 |
| Thrombin | S1/S2 | 6.50E-03 | 5.20E-02 | 16.34 | 3.18E-03 |
| Thrombin | S2′ | 7.34E-04 | 5.87E-03 | 13.98 | 4.20E-04 |

S2 cleavage at a rate intermediate between TMPRSS2 and factor Xa (**Figure 2D**). Unlike factor Xa, thrombin cleaves the S2′ peptide with greater activity than TMPRSS2 (**Figure 2D–F**).

We next compared the ability of coagulation factors to cleave SARS-CoV-2 S to their ability to cleave their known substrates. During the physiological process of clotting, factor Xa cleaves prothrombin at R271, which ultimately becomes the activated form α-thrombin (**Wood et al., 2011**). Thrombin, in turn, cleaves multiple sites of fibrinogen, including the beta chain (FGB) at R44, in a critical step toward aggregation and polymerization of high molecular weight fibrin clots. Fluorogenic peptides corresponding to THRB$^{R271}$ and FGB$^{R44}$ were synthesized and assayed with TMPRSS2, factor Xa, and thrombin. TMPRSS2 exhibited relatively broad activity to cleave this collection of substrates (**Figure 2G**). As expected, factor Xa showed strong selectivity for THRB$^{R271}$ over FGB$^{R44}$, while thrombin showed the opposite substrate preference (**Figure 2H–I**). Remarkably, factor Xa showed ~9-fold greater maximum initial reaction velocity ($V_{max}$) in cleaving the spike S1/S2 peptide compared to cleaving a peptide corresponding to its known substrate, THRB$^{R271}$ (**Figure 2H**). The $V_{max}$ for thrombin cleavage of the spike S1/S2 peptide was within ~4.5-fold of the $V_{max}$ for the benchmark FGB$^{R44}$ peptide (**Figure 2I**), indicating that thrombin might also cleave this site when activated during coagulation.

We next assessed the effect of substituting amino acids adjacent to the cleavage site of the S1/S2 peptide on proteolytic cleavage by these proteases. An arginine preceding the cleavage site (P1 position) is a common feature of substrates of many serine proteases. Substitution of the P1 arginine in the S1/S2 substrate with alanine (S1S2-P1A) resulted in a 4-fold reduction in TMPRSS2 cleavage and abolished nearly all cleavage by factor Xa and thrombin (**Figure 2—figure supplement 1A-C**). Substitutions in the P3 and P4 positions (S1S2-HPN) with features typical of a substrate of type II transmembrane serine proteases (TTSPs), a family which includes TMPRSS2 and hepsin (**Damalanka et al., 2019**), did not change TMPRSS2 cleavage and greatly reduced factor Xa and thrombin cleavage (**Figure 2—figure supplement 1A-C**). Although the substrate specificity of TTSPs and coagulation factors are not universally similar, the cleavage sites of SARS-CoV-2 are specifically cleaved by TMPRSS2, factor Xa, and thrombin.

To consider whether the context of protease activity may influence substrate specificity, we repeated the enzymatic assay in the presence of phospholipids. The addition of phospholipid vesicles did not change factor Xa activity or substrate preference in this assay (**Figure 2—figure supplement 1E**). To ensure the quality of our phospholipid vesicle preparation, we added 0–100 µM PC/PS to a dilute Russell's viper venom time (dRVVT) clotting assay, where PC/PS drives a significant, concentration-dependent acceleration of clotting of normal pooled human plasma (**Figure 2—figure supplement 1F**). In summary, the coagulation serine proteases factor Xa and thrombin exhibit proteolytic activity against SARS-CoV-2 spike peptide substrates.

## Factor Xa and thrombin facilitate SARS-CoV-2 spike-mediated entry

We next investigated whether coagulation factors could cleave trimeric spike in its native 3D conformation, and whether this activity potentiated spike function in viral entry into cells. To do so, we used replication-defective SARS-CoV-2 spike-pseudotyped VSV or HIV-1 virus (**Schmidt et al., 2020**). In the VSV pseudovirus system, addition of purified factor Xa or thrombin to the media significantly

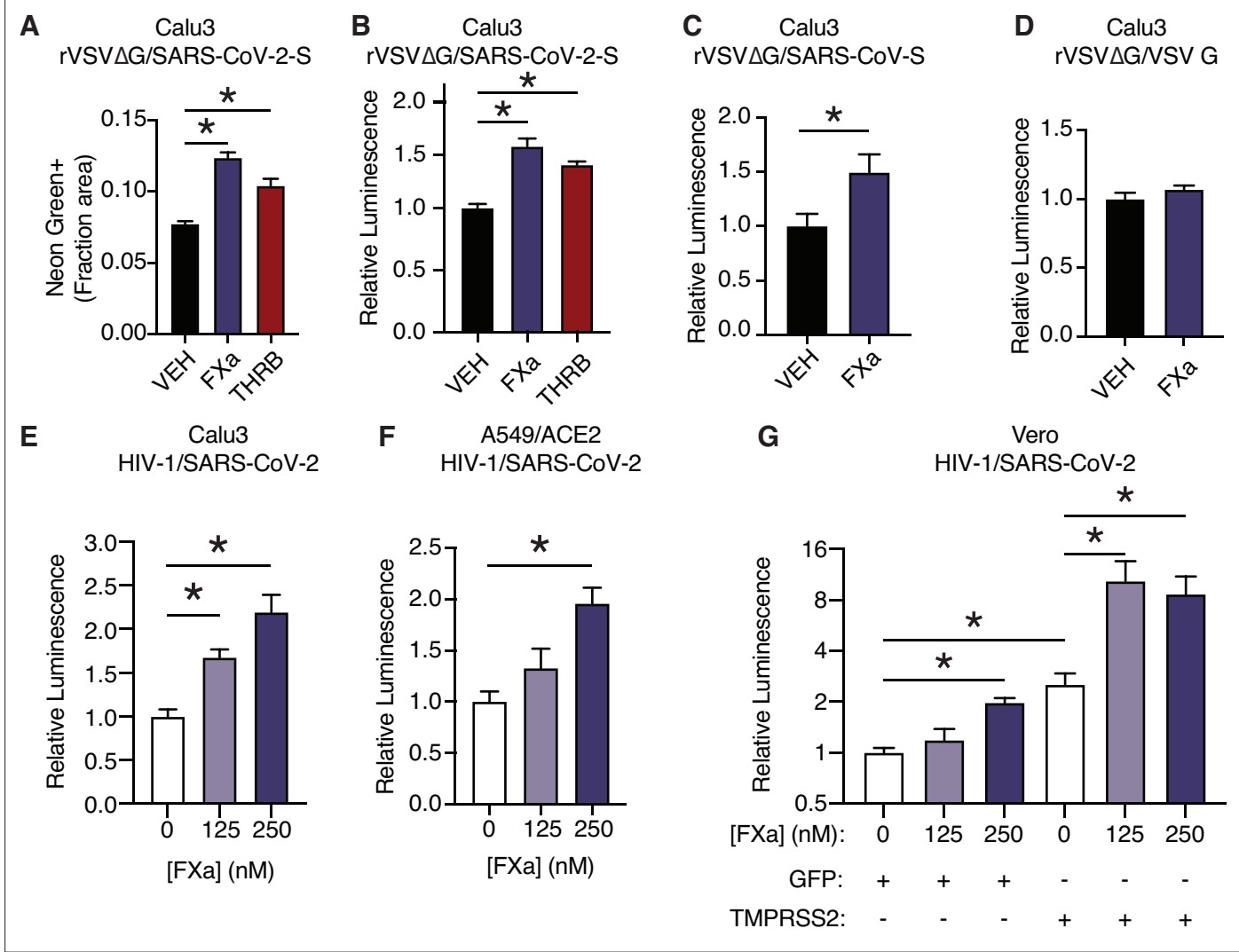

**Figure 3.** Factor Xa and thrombin facilitate SARS-CoV-2 spike-mediated entry. (**A**) Calu3 cells were infected with rVSVΔG/SARS-CoV-2 pseudovirus with concomitant treatment with vehicle, 250 nM factor Xa, or 250 nM thrombin. Quantification of the ratio of green fluorescent area to total confluence (4 fields/replicate well, 4 wells/condition). (**B**) Nanoluciferase luminescent signal was measured following infection with rVSVΔG/SARS-CoV-2 pseudovirus and the addition of either vehicle, factor Xa, or thrombin. The effect of factor Xa on rVSVΔG complemented with either (**C**) SARS-CoV spike or (**D**) VSV-G was measured by luminescent signal. Luminescent signal was measured following HIV-1$_{NL}$/SARS-CoV-2 pseudovirus infection and concomitant treatment with 125–250 nM factor Xa in (**E**) Calu3 cells, (**F**) A549/ACE2, and (**G**) Vero cells following transduction with lentiviral vectors to express GFP or TMPRSS2. Following selection, cells were infected with HIV-1$_{NL}$/SARS-CoV-2 pseudovirus and concomitantly treated with 125–250 nM factor Xa. Subsequently, nanoluciferase luminescent signal was determined and plotted relative to vehicle-treated control. *$p < 0.05$, two-tailed t-test. Data represented as mean ± SEM.

The online version of this article includes the following source data and figure supplement(s) for figure 3:

**Source data 1.** Data and summary statistics for pseudovirus assays with exogenous proteases.

**Figure supplement 1.** Further characterization of coagulation factor-induced SARS-CoV-2 pseudovirus infection.

**Figure supplement 2.** Assessing relevant protease levels with ex vivo clotting assays.

increased infection in Calu3 cells 16 hr post infection as determined by either quantification of Neon-Green (*Figure 3A*, *Figure 3—figure supplement 1A*) or nanoluciferase activity (*Figure 3B*).

While factor X and prothrombin levels are variable between individuals in healthy populations (*Brummel-Ziedins et al., 2012*), the concentration of proteases used in the pseudovirus assay (125–250 nM) are comparable to reference ranges of factor X (*Brummel-Ziedins et al., 2012*; *Tormoen et al., 2013*; *Williams and Marks, 1994*) and prothrombin (*Baugh et al., 1998*; *Brummel-Ziedins et al., 2012*; *Tormoen et al., 2013*). Similar concentrations of active purified proteases were required

to normalize in vitro clotting times, where purified factor Xa was used to correct dRVVT of factor X-deficient human plasma (*Figure 3A*, *Figure 3—figure supplement 2A*) and purified thrombin was used to correct the prothrombin time of prothrombin-deficient human plasma (*Figure 3—figure supplement 2B*).

SARS-CoV-2 contains a notable insertion of basic residues at the S1/S2 boundary, distinguishing its sequence from many related betacoronaviruses (*Jaimes et al., 2020a*). Entry of rVSVΔG was increased when complemented with spike protein from SARS-CoV of the 2002 outbreak (*Figure 3C*), but not when complemented instead with VSV G (*Figure 3D*). This indicates that factor Xa spike cleavage could be relevant across multiple coronaviruses, but is not generally associated with VSV entry.

We further validated that factor Xa activated spike-mediated entry the HIV-1-based pseudovirus system (*Schmidt et al., 2020*). Consistent with the results above, addition of purified factor Xa to the media at the time of infection enhanced entry of HIV-1-based SARS-CoV-2 pseudovirus in Calu3 cells (*Figure 3E*). Thrombin did not appear to enhance spike-mediated entry by HIV-1/SARS-CoV-2 pseudovirus, unlike rVSVΔG/SARS-CoV-2 pseudovirus (*Figure 3—figure supplement 1B-E*).

We investigated the functional interplay of TMPRSS2 expression and the effect of exogenous activated coagulation factors. TMPRSS2 is expressed in Calu3 cells and contributes to coronavirus entry (*Hoffmann et al., 2020a*), whereas A549/ACE2 and Vero cells lack endogenous TMPRSS2 expression. Factor Xa induced a significant dose-dependent effect on pseudovirus entry in both Calu3 and A549/ACE2 (*Figure 3E–F*). Furthermore, an isogenic pair of Vero cells was generated by expressing TMPRSS2 or GFP control. Pseudovirus infection of both Vero$^{GFP}$ and Vero$^{TMPRSS2}$ cells were significantly increased by factor Xa, indicating that factor Xa enhancement of infection is not dependent on TMPRSS2 (*Figure 3G*). This is consistent with the model that FXa cuts the S1/S2 site and TMPRSS2 has functionally important activity at the S2' site.

## Nafamostat broadly inhibits cleavage of spike peptides by both transmembrane serine proteases and coagulation factors

Given that multiple proteases could contribute to SARS-CoV-2 spike cleavage activation, a drug that could inhibit both transmembrane serine proteases and coagulation factors would be potentially valuable as a dual action anticoagulant/antiviral in COVID-19. Accordingly, we next explored the candidate set of inhibitors for cross-reactivity against a broader set of proteases that could facilitate viral entry. Like TMPRSS2, human airway trypsin-like protease (HAT), encoded by *TMPRSS11D*, is a member of the TTSP family of proteases and could activate SARS-CoV-2 S (*Hoffmann et al., 2021*). HAT exhibited sensitivity to camostat and nafamostat, similar to TMPRSS2 (*Figure 4A–B*). Compared to TMPRSS2, HAT was more sensitive to dabigatran and less sensitive to otamixaban (*Figure 4A–B*). Factor Xa activity against the S1/S2 peptide was most sensitive to otamixaban and moderately sensitive to nafamostat and dabigatran (*Figure 4C*). Thrombin activity was sensitive to camostat, nafamostat, and dabigatran, and moderately sensitive to otamixaban (*Figure 4D*).

We performed a dose-response curve of the panel of inhibitors on factor Xa and thrombin S1/S2 cleavage. Otamixaban, a designed factor Xa inhibitor, demonstrated an IC50 at the nanomolar level to factor Xa, while nafamostat and dabigatran demonstrated IC50s in the micromolar range (*Figure 4E*). Camostat did not potently inhibit factor Xa spike cleavage. Dabigatran, a designed thrombin inhibitor, as well as nafamostat and camostat demonstrated a submicromolar IC50 for thrombin-dependent spike cleavage (*Figure 4F*). Otamixaban inhibited thrombin spike cleavage in the micromolar range.

Furin showed high activity against the S1/S2 peptide, but not against the S2' peptide, and was not sensitive to any of the candidate inhibitors (*Figure 4—figure supplement 1A-B*). While it has been suggested that TMPRSS4 or neutrophil elastase may also cleave SARS-CoV-2 spike, we detected minimal activity against either S1/S2 or S2' peptide substrates in our enzymatic assay (<1% of furin cleavage of S1/S2) (*Figure 4—figure supplement 1A-B*).

In the pseudovirus assay, nafamostat effectively suppresses SARS-CoV-2 S-mediated entry with or without the addition of exogenous factor Xa, using either the VSV-based (*Figure 4G*) or HIV-1-based (*Figure 4H*) SARS-CoV-2 pseudovirus. To clarify the pleiotropic nature of nafamostat, which inhibits TMPRSS2 and factor Xa, we compared the effect of apixaban, which inhibits factor Xa but not TMPRSS2. Apixaban rescued the effect of exogenous FXa back to the baseline level of infection, but did not affect pseudovirus infection in the absence of exogenous protease (*Figure 4—figure supplement 2*). Direct oral anticoagulants (DOACs) have the potential to block clotting factor-mediated

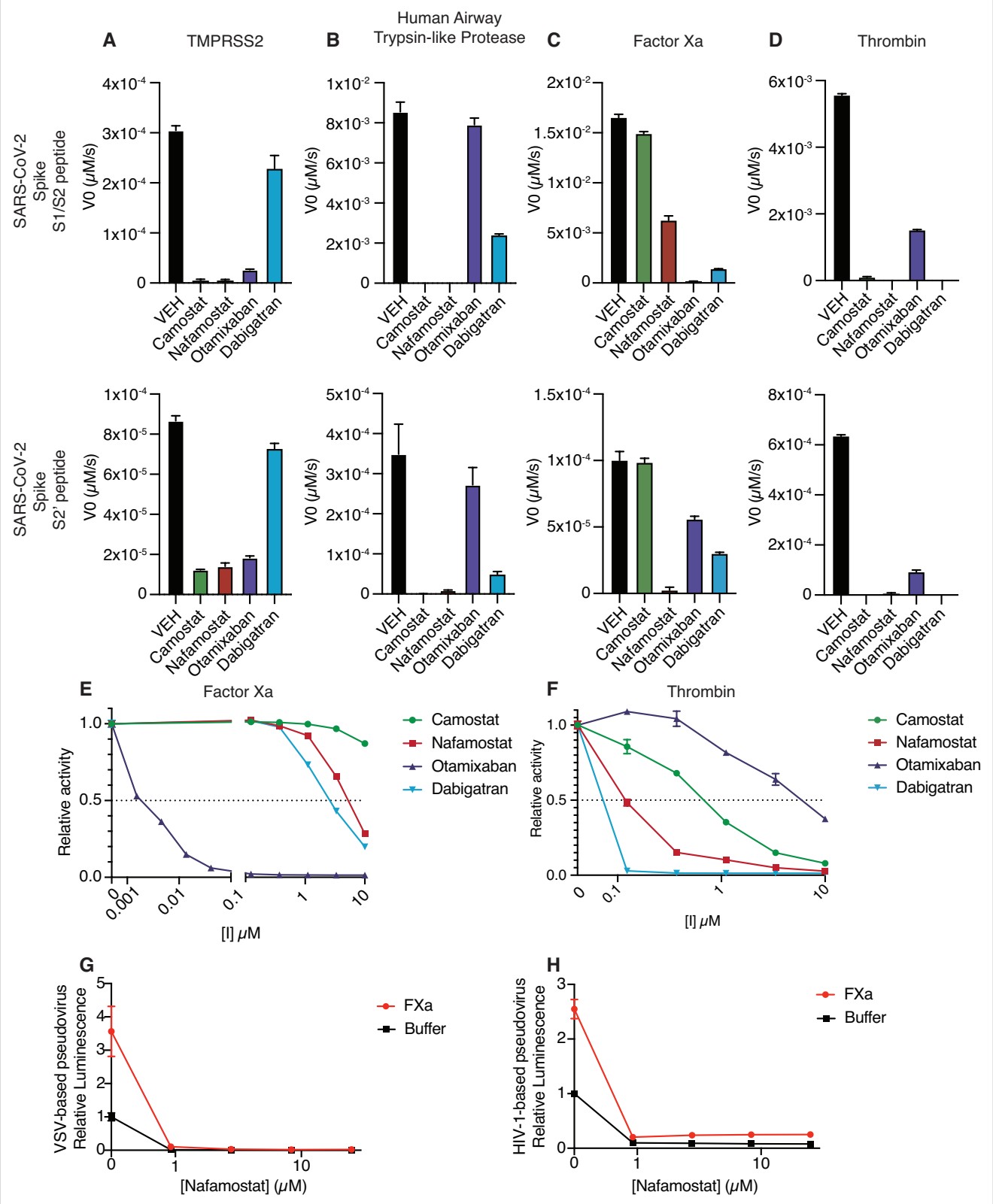

**Figure 4.** Nafamostat broadly inhibits cleavage of spike peptides by both coagulation factors and transmembrane serine proteases. Initial velocities for the cleavage of 10 μM SARS-CoV-2 spike S1/S2 (top) and S2' (bottom) peptide substrates by (**A**) TMPRSS2, (**B**) TMPRSS11D/human airway trypsin-like protease (**C**) factor Xa, and (**D**) thrombin were measured in the presence of DMSO vehicle, or 10 μM camostat, nafamostat, otamixaban, or dabigatran. The relative activity of (**E**) factor Xa and (**F**) thrombin were determined over a range of 0–10 μM of the indicated drugs. Calu3 cells were treated with a

*Figure 4 continued on next page*

*Figure 4 continued*

range of concentrations of nafamostat with or without addition of 250 nM exogenous factor Xa and infected with (**G**) rVSVΔG/SARS-CoV-2 pseudovirus or (**H**) HIV-1$_{NL}$/SARS-CoV-2 pseudovirus and infectivity was measured by luminescence. N = 3, data represented as mean ± SEM.

The online version of this article includes the following source data and figure supplement(s) for figure 4:

**Source data 1.** Data and summary statistics for enzymatic assays to determine the effects of protease inhibitors on host proteases.

**Figure supplement 1.** Activity of candidate inhibitors against other proteases.

**Figure supplement 2.** Apixaban rescues effect of factor Xa, related to *Figure 4*.

enhancement of viral entry, but TMPRSS2-mediated cleavage activation would remain unaffected by treatment with DOACs in common usage in North America and Europe. Taken together, nafamostat appears to be a versatile inhibitor of spike activation by a variety of TTSPs and coagulation factors. The multitarget mechanism of nafamostat distinguishes its potential as an antiviral/anticoagulant from currently FDA-approved DOACs.

## Factor Xa and thrombin increase SARS-CoV-2 infection in lung organoids

To explore the effect of coagulation factors in a more physiologically relevant setting, we exposed human pluripotent stem cell-derived lung organoids (hPSC-LOs) to SARS-CoV-2 infection under BSL-3 conditions with or without addition of exogenous protease. Stepwise, directed differentiation of human pluripotent stem cells generates lung organoids that form three-dimensional cellular structures and recapitulate functional and molecular characteristics of the lung (*Chen et al., 2019*; *Chen et al., 2017*; *Han et al., 2021*; *Huang et al., 2014*; *Mou et al., 2012*). hPSC-LOs express alveolar type II cell markers and were previously shown to be permissive to SARS-CoV-2 infection and replication (*Han et al., 2021*).

Mature lung organoids were infected with SARS-CoV-2 at a multiplicity of infection (MOI) of 0.1 (*Figure 5A–B*) or 0.01 (*Figure 5C–D*) and viral replication was measured by quantitative real-time PCR (qRT-PCR) for SARS-CoV-2 nucleocapsid (*N*) (*Figure 5A and C*) and SARS-CoV-2 envelope (*E*) (*Figure 5B and D*). The addition of either factor Xa or thrombin increased the levels of viral RNA following infection (*Figure 5A–D*). This effect was accentuated at 48 hr post infection with respect to 24 hr post infection, and with lower initiating MOI. Factor Xa and thrombin increase SARS-CoV-2 infection in the context of multicycle viral replication in human lung organoids.

## Discussion

### Coagulation factors cleave the SARS-CoV-2 spike protein

Using a FRET-based enzymatic assay, two platforms of pseudovirus assays, and SARS-CoV-2 infection experiments in lung organoids, we demonstrate that coagulation proteases factor Xa and thrombin cleave SARS-CoV-2 spike protein. Coagulation-induced cleavage enhances spike activation and increases viral entry into target cells, potentially instigating a positive feedback loop with infection-induced coagulation. Nafamostat, among currently available drugs, is best suited as a multi-purpose inhibitor against spike cleavage by TTSPs and coagulation factors. These data have numerous implications at the intersection of virology and coagulation.

### Viral envelope protein activation by non-target cell proteases

Hijacking of host transmembrane, endosomal, and ER proteases to activate viral envelope proteins has been described in influenza A, human metapneumovirus, HIV, and Sendai virus (*Kido et al., 1996*; *Straus et al., 2020*). In the present study, we find an instance where the virus can be primed not by proteases expressed by the target cell, but by host organism proteases derived from the microenvironment of the target cell. Prior studies have described cleavage activation of SARS-CoV by elastase and plasmin, illustrating that microenvironmental host proteases can indeed play an important role in coronavirus spike priming (*Belouzard et al., 2010*; *Kam et al., 2009*; *Matsuyama et al., 2005*). Generally, the scope by which circulating proteases, such as coagulation factors, or proteases expressed by immune cells interact with viral envelope proteins during infection has not been comprehensively explored.

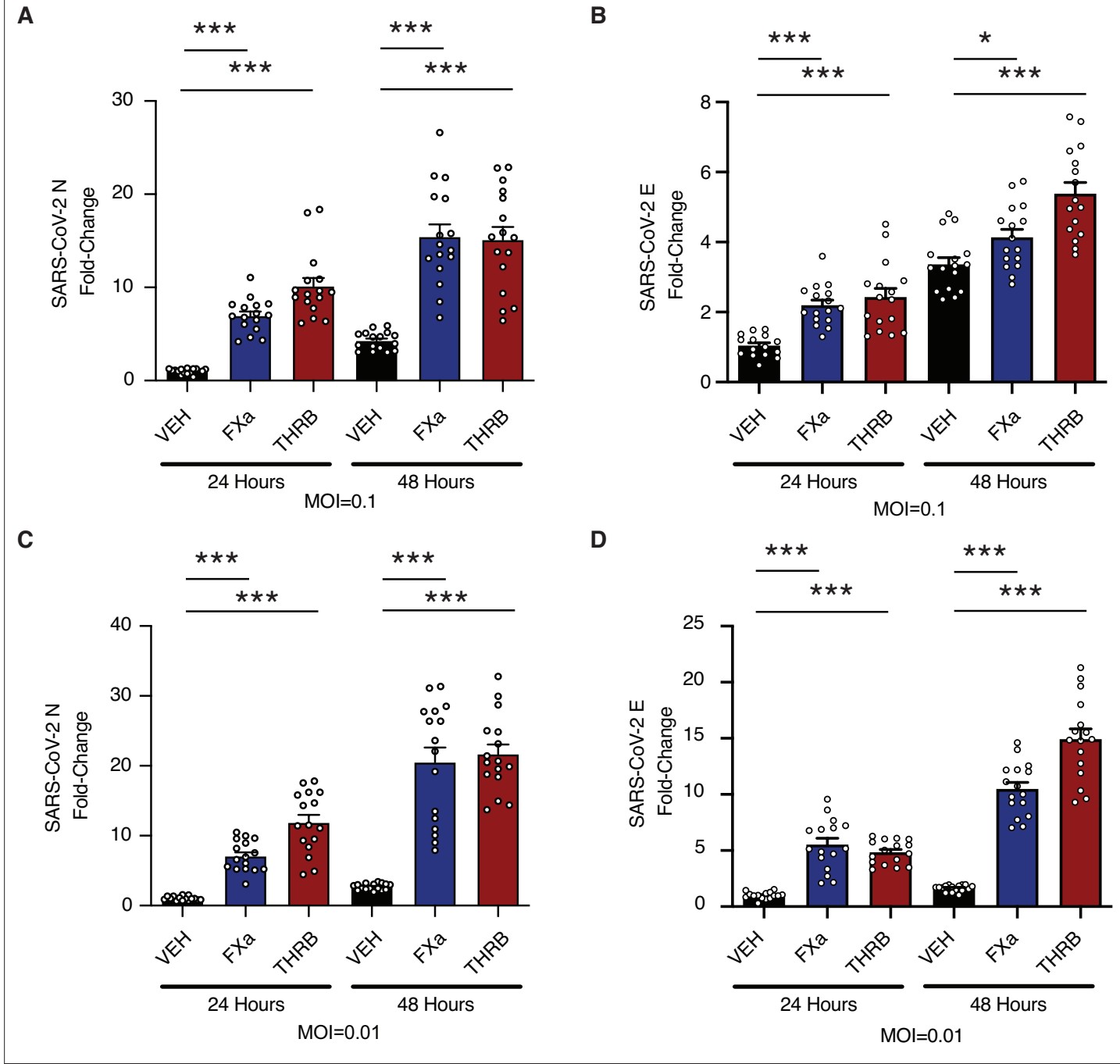

**Figure 5.** Factor Xa and thrombin increase SARS-CoV-2 infection in lung organoids. Human pluripotent stem cell (hPSC)-derived lung organoids were infected with SARS-CoV-2. Upon infection, organoids were treated with 170 nM of purified factor Xa or thrombin. (**A**) Relative level of SARS-CoV-2-N RNA following infection at multiplicity of infection (MOI) = 0.1. (**B**) Relative level of SARS-CoV-2-E RNA following infection at MOI = 0.1. (**C**) Relative level of SARS-CoV-2-N RNA following infection at MOI = 0.01. (**D**) Relative level of SARS-CoV-2-E RNA following infection at MOI = 0.01. N = 16, *p = 0.0144, ***p < 0.0001, two-tailed t-test. Data represented as mean ± SEM.

The online version of this article includes the following source data for figure 5:

**Source data 1.** Data and summary statistics for infection assays with exogenous proteases.

Relatively few studies have examined the interaction of factor Xa or thrombin and viral proteins, and each relies on target cells as the source of coagulation factors. Our results are consistent with a prior study that concluded that factor Xa cleaves and activates SARS-CoV spike (*Du et al., 2007*). Traditionally, influenza vaccines rely on viral propagation in chicken eggs, where hemagglutinin cleavability by factor Xa is a determinant of the efficiency of strain-specific propagation of influenza A virus in ovo (*Gotoh et al., 1990*; *Straus and Whittaker, 2017*). Hepatitis E virus ORF1 polyprotein is processed intracellularly by thrombin and factor Xa in the cytoplasm of hepatocytes, which are the primary cell type responsible for generating and secreting coagulation factors (*Kanade et al., 2018*).

Activation of coagulation has the potential to exacerbate SARS-CoV-2 infectivity in both TMPRSS2$^+$ and TMPRSS2$^-$ host cells. Reliance on extracellular proteolytic activity could expand the field of susceptible cell types and regions of the airway. Extrapulmonary infection has been described, particularly in small intestinal enterocytes (*Xiao et al., 2020*; *Zang et al., 2020*) and, in some cases, the central nervous system (*Song et al., 2021*). It warrants investigation whether hypercoagulation is linked to extrapulmonary infection.

## Evolutionary perspective on viral-host interaction

Proteolytic cleavage of the spike forms a barrier to zoonotic crossover independent of receptor binding (*Menachery et al., 2020*). Hemostasis is of central importance in mammals and represents a major vulnerability of mammals to predators and pathogens, either through hyperactivation of coagulation or uncontrolled bleeding. The dysregulation of hemostasis is a convergent mechanism of toxins of snakes, bees, and bats (*Ma et al., 2013*; *Markland and Swenson, 2013*; *Prado et al., 2010*) and a driver of virulence in Ebola and dengue virus infection (*Geisbert et al., 2003*; *Rathore et al., 2019*). Acute lung injury from viral cytopathic effects, the induction of a COVID-19-associated cytokine storm, complement activation, and anti-phospholipid autoantibodies have all been suggested to instigate the coagulation cascade (*Merrill et al., 2020*; *Zuo et al., 2020*). Furthermore, one model posits that COVID-19 coagulopathy is platelet-driven and an Arg-Gly-Asp (RGD) motif on SARS-CoV-2 spike directly interacts with GPIIb/GPIIIa integrins on the surface of platelets (*Cox, 2021*), consistent with in silico predictions of integrin binding (*Mészáros et al., 2021*). Perhaps, SARS-CoV-2 has undergone selection to exploit an environment locally enriched in coagulation proteases for enhanced entry. As infection spreads, more clotting is induced, instigating a positive feedback loop to promote entry into additional host cells.

## Clinical relevance of potential antiviral activity of anticoagulants

Effective anticoagulation is a critical area of investigation to improve outcomes in coronavirus infection. Vitamin K antagonists, including heparin, are commonly used for preventing venous thromboembolism in COVID-19, although no strong evidence yet supports any specific anticoagulant (*Cuker et al., 2021*). Three large randomized clinical trials to determine the benefit of therapeutic intensity vs. prophylactic intensity heparin in critically ill COVID-19 patients were suspended at interim analysis for futility (NCT02735707, NCT04505774, and NCT04372589). There has been interest in the use of direct-acting oral anticoagulants (DOACs) to manage COVID-19-related coagulopathy, but optimal protocols for managing coagulopathy in COVID-19 patients have not yet been developed (*Capell et al., 2021*; *Lopes et al., 2021*). The most prominent risk of anticoagulants is bleeding, and notably DOACs, as well as nafamostat, have a reduced risk of intracranial hemorrhage and other bleeding events compared to vitamin K antagonists (*Chen et al., 2020*; *Hellenbart et al., 2017*; *Makino et al., 2016*).

In our studies, anticoagulant serine protease inhibitors, otamixaban and dabigatran, exhibited off-target activity against TMPRSS2 and other TTSPs, but likely require concentrations higher than those safely reached in vivo (*Paccaly et al., 2006*; *Stangier and Clemens, 2009*). On the other hand, our data suggest that nafamostat and camostat may offer three distinct therapeutic mechanisms against SARS-CoV-2 infection; these compounds have the potential to block spike cleavage mediated by TMPRSS2 and other TTSPs, to block spike cleavage by coagulation factors, and to serve as an anticoagulant. It is also plausible that nafamostat, like related protease inhibitor pentamidine, could also interfere with platelet GPIIb/IIIa, platelet aggregation, and thrombus formation (*Cox, 2021*; *Cox et al., 1996*). Nafamostat (*Fujii and Hitomi, 1981*; *Keck et al., 2001*; *Takeda et al., 1996*) and Camostat (*Ramsey et al., 2019*) have been in clinical use in Asia for many years for the treatment of pancreatitis.

Nafamostat has also been used as an anticoagulant during hemodialysis (*Akizawa et al., 1993*) and extracorporeal membrane oxygenation (*Park et al., 2015*), and to manage disseminated intravascular coagulopathy (*Kobayashi et al., 2001*). As of this writing, there have been eight clinical trials initiated (reported on https://www.clinicaltrials.gov/) to investigate the use of nafamostat in COVID-19, while 24 active clinical trials of camostat for COVID-19 were identified.

Inhibition of coagulation factor-induced spike cleavage may contribute to the molecular mechanism of these agents, if treatment is given sufficiently early. Many COVID-19-associated complications leading to hospitalization occur as immune hyperactivation waxes and peak viral titer wanes (*Griffin et al., 2021*). To take advantage of the potential antiviral effect of anticoagulants, early intervention in an outpatient setting may be beneficial.

## Limitations

The experiments of this study, like prior studies using similar techniques, have some limitations. Protease enzymatic assays on peptide substrates allow for detailed biochemical characterization of a specific site, but peptide substrates may not have the equivalent three-dimensional conformation or post-translational modifications of the full-length protein produced in appropriate cells. For instance, SARS-CoV-2 S is extensively glycosylated (*Watanabe et al., 2020*). The possibility of additional spike cleavage sites and potential pro- and anti-viral consequence of proteases acting on cell surface proteins including ACE2 cannot be excluded. The amount, density, and accessibility of spike protein could be different between pseudovirus assays and wild type (WT) SARS-CoV-2 infection. However, antibody neutralization is highly correlated between authentic virus and corresponding pseudotyped viruses, suggesting similar conformation (*Schmidt et al., 2020*). We confirmed our results using WT SARS-CoV-2 infection, which alone does not easily allow for precise definition of which stage of the viral replication cycle is being affected, but pseudovirus assays confirm a cell-entry mechanism. We have attempted to mitigate the risk of artifact by using multiple orthogonal platforms.

## Conclusion

Collectively, our data provide rationale for the investigation of early intervention with judiciously selected anticoagulant treatment, which may have collateral benefit in limiting progressive spread of SARS-CoV-2 infection throughout the lung in infected individuals. Preparedness to mitigate future coronavirus outbreaks is critical to pursue through the understanding of coronavirus-host interactions.

# Materials and methods

**Key resources table**

| Reagent type (species) or resource | Designation | Source or reference | Identifiers | Additional information |
|---|---|---|---|---|
| Chemical compound, drug | Camostat | Selleck | Cat# S2874 | |
| Chemical compound, drug | Nafamostat | Selleck | Cat# S1386 | |
| Chemical compound, drug | Apixaban | Medchem Express | Cat# HY-50667 | |
| Chemical compound, drug | Betrixaban | Medchem Express | Cat# HY-10268 | |
| Chemical compound, drug | Bivalirudin (TFA) | Medchem Express | Cat# HY-15664 | |
| Chemical compound, drug | Boceprevir | Medchem Express | Cat# HY-10237 | |
| Chemical compound, drug | Dabigatran etexilate | Medchem Express | Cat# HY-10274 | |
| Chemical compound, drug | Edoxaban | Medchem Express | Cat# HY-10264 | |
| Chemical compound, drug | Otamixaban | Medchem Express | Cat# HY-70035 | |
| Chemical compound, drug | Rivaroxaban | Medchem Express | Cat# HY-50903 | |
| Chemical compound, drug | Simeprevir | Medchem Express | Cat# HY-10241 | |
| Chemical compound, drug | Sivelestat | Medchem Express | Cat# HY-17443 | |
| Chemical compound, drug | Telaprevir | Medchem Express | Cat# HY-10235 | |
| Chemical compound, drug | Dabigatran | Medchem Express | Cat# HY-10163 | |

*Continued on next page*

*Continued*

| Reagent type (species) or resource | Designation | Source or reference | Identifiers | Additional information |
|---|---|---|---|---|
| Peptide, recombinant protein | Thrombin | Millipore Sigma | Cat# 605195 | |
| Peptide, recombinant protein | Factor Xa | Millipore Sigma | Cat# 69036 | |
| Peptide, recombinant protein | TMPRSS2 | LSBio | Cat# LS-G57269 | |
| Peptide, recombinant protein | TMPRSS4 | Aviva System Biology | Cat# OPCA0240 | |
| Peptide, recombinant protein | Furin | Thermo Fisher Scientific | Cat# 1503SE010 | |
| Peptide, recombinant protein | Neutrophil elastase | Thermo Fisher Scientific | Cat# 9167SE020 | |
| Peptide, recombinant protein | S1/S2 | Anaspec | | QXL520-PRRARSVASQ-K(5-FAM)-NH2 |
| Peptide, recombinant protein | S2' | Anaspec | | QXL520-KPSKRSFIED-K(5-FAM)-NH2 |
| Peptide, recombinant protein | THRB-R271 | Anaspec | | QXL520-AIEGRTATSE-K(5-FAM)-NH2 |
| Peptide, recombinant protein | FGB-R44 | Anaspec | | QXL520-FFSARGHRPL-K(5-FAM)-NH2 |
| Peptide, recombinant protein | S1/S2-P1A | Anaspec | | QXL520-PRRAASVASQ-K(5-FAM)-NH2 |
| Peptide, recombinant protein | S1/S2-HPN | Anaspec | | QXL520-PSQARSVASQ-K(5-FAM)-NH2 |
| Chemical compound, drug | Phosphatidylcholine | Avanti Polar Lipids | Cat# 850375C | 1,2-Dioleoyl-*sn*-glycero-3-phosphocholine |
| Chemical compound, drug | phosphatidylserine | Avanti Polar Lipids | Cat# 840035C | 1,2-Dioleoyl-*sn*-glycero-3-phospho-L-serine |
| Cell line (*Homo sapiens*) | Calu3 | ATCC | Cat# HTB-55; RRID:CVCL_0609 | |
| Cell line (*Homo sapiens*) | A549 | ATCC | Cat# CCL-185; RRID:CVCL_0023 | |
| Cell line (*Homo sapiens*) | Caco2 | ATCC | Cat# HTB-37; RRID:CVCL_0025 | |
| Cell line (*Chlorocebus sabaeus*) | Vero | Laboratory of Benjamin tenOever | RRID:CVCL_0059 | |
| Cell line (*Homo sapiens*) | HEK293T | ATCC | Cat# CRL-3216; RRID:CVCL_0063 | |
| Recombinant DNA reagent | pEGPN | This paper | | |
| Recombinant DNA reagent | pEGPN-ACE2 | This paper | | |
| Recombinant DNA reagent | pEGPN-TMPRSS2 | This paper | | |
| Recombinant DNA reagent | Lenti-Cas9-blast | Addgene | Cat# 52962 | |
| Recombinant DNA reagent | ipUSEPR | Francisco Sanchez-Rivera & Scott Lowe | | |
| Recombinant DNA reagent | CMV-SARS-CoV-2-S | *Schmidt et al., 2020* | | |
| Recombinant DNA reagent | CCNanoLuc/GFP | *Schmidt et al., 2020* | | |
| Recombinant DNA reagent | HIV-1NL GagPol | *Schmidt et al., 2020* | | |
| Commercial assay or kit | NEBuilder master mix | New England Biolabs | Cat# E2621 | |
| Chemical compound, drug | XtremeGene9 | Millipore Sigma | Cat# 6365787001 | |
| Chemical compound, drug | Polybrene | Santa Cruz Biotechnology | Cat# SC-134220 | |
| Chemical compound, drug | Lenti-X | Takara Bio | Cat# 631232 | |
| Chemical compound, drug | G418 | Sigma-Aldrich | Cat# # G8168 | |
| Chemical compound, drug | Blasticidin | Invivogen | Cat# ANT-BL-1 | |
| Chemical compound, drug | Puromycin | Thermo Fisher Scientific | Cat# A1113803 | |
| Commercial assay or kit | Cell Lysis Buffer | Promega | Cat# E1531 | |
| Commercial assay or kit | NanoGlo Luciferase Assay | Promega | Cat# N1130 | |

*Continued on next page*

*Continued*

| Reagent type (species) or resource | Designation | Source or reference | Identifiers | Additional information |
|---|---|---|---|---|
| Biological sample (*Homo sapiens*) | Normal human plasma | Pacific Hemostasis | Cat# 95059–698 | |
| Biological sample (*Homo sapiens*) | Factor X-deficient plasma | Haematologic Technologies | Cat# FX-ID | |
| Biological sample (*Homo sapiens*) | Prothrombin-deficient plasma | Haematologic Technologies | Cat# FII-ID | |
| Biological sample (*Vipera russelli*) | Russell's Viper Venom | Sigma-Aldrich | Cat# V2501 | |
| Strain, strain background (*Indiana vesiculovirus*) | rVSVdG/NG-NanoLuc | *Schmidt et al., 2020* | | |
| Strain, strain background (SARS-CoV-2) | SARS-CoV-2, isolate USA-WA1/2020 | BEI Resources, NIAID, NIH | Cat# NR-52281 | |
| Sequence-based reagent | sgRNA: CD4 | This study | sgRNA | GGTGCAATGTAGGAGTCCAA |
| Sequence-based reagent | sgRNA: PHGDH intron1 | This study | sgRNA | GGGCGAGAGAGAGAAAATTG |
| Sequence-based reagent | sgRNA: ACE2 g1 | This study | sgRNA | CACCGCAAAGGCGAGAGATAGTTG |
| Sequence-based reagent | sgRNA: ACE2 g2 | This study | sgRNA | CACCGACATCTTCATGCCTATGTG |
| Sequence-based reagent | sgRNA: TMPRSS2 g1 | This study | sgRNA | CACCGCTGGAACGAGAACTACGGG |
| Sequence-based reagent | sgRNA: TMPRSS2 g2 | This study | sgRNA | CACCGGGGACGGGTAGTACTGAGC |
| Sequence-based reagent | Primer: CD4-Forward | This study | PCR primers | GATAATGGAGAGATGTTGTTGGTTT |
| Sequence-based reagent | Primer: CD4- Reverse | This study | PCR primers | ATGTCCAGGTGCCACTATCCT |
| Sequence-based reagent | Primer: PHGDH intron 1 – Forward | This study | PCR primers | AAAGCAGAACCTTAGCAAAGAGG |
| Sequence-based reagent | Primer: PHGDH intron 1 – Reverse | This study | PCR primers | GAACTAATTGATACGGGGTGCAT |
| Sequence-based reagent | Primer: ACE2-g1- Forward | This study | PCR primers | TCCCTACTTTTTGTCGTTATTAGCA |
| Sequence-based reagent | Primer: ACE2-g1- Reverse | This study | PCR primers | GGTGATCCACAGCTAATGTATTGTT |
| Sequence-based reagent | Primer: ACE2-g2- Forward | This study | PCR primers | TCAAAATGCGATTTCTACAATGTTA |
| Sequence-based reagent | Primer: ACE2-g2- Reverse | This study | PCR primers | TGGGCTTTTCAGATTAAACCATTAT |
| Sequence-based reagent | Primer: TMPRSS2-g1-Forward | This study | PCR primers | ACAAATTCCACCTGCTGGTTATAG |
| Sequence-based reagent | Primer: TMPRSS2-g1- Reverse | This study | PCR primers | ACTTCATCCTTCAGGTGTACTCATC |
| Sequence-based reagent | Primer: TMPRSS2-g2- Forward | This study | PCR primers | CAGGAAATAAACACAAAGAGAATCC |
| Sequence-based reagent | Primer: TMPRSS2-g2-Reverse | This study | PCR primers | ACTATGAAAACCATGGATACCAACC |
| Sequence-based reagent | SARS-CoV-2-N-F | | PCR primers | TAATCAGACAAGGAACTGATTA |
| Sequence-based reagent | SARS-CoV-2-N-R | | PCR primers | CGAAGGTGTGACTTCCATG |
| Sequence-based reagent | SARS-CoV-2-E-F | | PCR primers | ACAGGTACGTTAATAGTTAATAGCGT |
| Sequence-based reagent | SARS-CoV-2-E-R | | PCR primers | ATATTGCAGCAGTACGCACACA |
| Sequence-based reagent | Human 18S-F | | PCR primers | GGCCCTGTAATTGGAATGAGTC |
| Sequence-based reagent | Human 18S-R | | PCR primers | CCAAGATCCAACTACGAGCTT |
| Software, algorithm | Prism 9 | GraphPad Software | | |

## Enzymatic assay

Thrombin (605195, human, activated by factor Xa, factor Va, and phospholipid) and factor Xa (69036, bovine, activated by Russell's Viper Venom) were obtained from Millipore Sigma. Recombinant TMPRSS2, purified from yeast, was obtained from LSBio (LS-G57269). TMPRSS4 was obtained from Aviva System Biology (OPCA0240), furin was obtained from Thermo Fisher Scientific (1503SE010), neutrophil elastase was obtained from Thermo Fisher Scientific (9167SE020). FRET peptides were

obtained from Anaspec, and a peptide sequences are listed in *Figure 2—figure supplement 1D*. Protease assay buffer was composed of 50 mM Tris-HCl, 150 mM NaCl, pH 8. Enzyme dilution/storage buffer was 20 mM Tris-HCl, 500 mM NaCl, 2 mM CaCl$_2$, 50% glycerol, pH 8. Peptides were reconstituted and diluted in DMSO. Enzyme kinetics were assayed in black 96-well plates with clear bottom and measured using a BMG Labtech FLUOstar Omega plate reader, reading fluorescence (excitation 485 nm, emission 520 nm) every minute for 20 cycles, followed by every 5 min for an additional eight cycles. A standard curve of 5-FAM from 0 to 10 µM (1:2 serial dilutions) was used to convert RFU to µM of cleaved FRET peptide product. Calculation of enzyme constants was performed with GraphPad Prism software (version 9.0). Camostat and nafamostat were obtained from Selleck Chemicals and all other inhibitors were obtained from MedChem Express.

## Phospholipid vesicles

Phosphatidylcholine (1,2-dioleoyl-*sn*-glycero-3-phosphocholine, Avanti Polar Lipids #850375C) and phosphatidylserine (1,2-dioleoyl-*sn*-glycero-3-phospho-L-serine, Avanti Polar Lipids #840035C) were mixed in a 3:1 w/w ratio in chloroform solvent in a screw top vial and the chloroform solvent was evaporated under a nitrogen stream. Unilamellar vesicles were isolated by extrusion using 0.1 µm pore filters and diluted in buffer AB2 (50 mM Tris-HCl, 150 mM NaCl, pH 8).

## Cell culture

Calu3, A549, Caco2, and Vero cells were tested for mycoplasma (Lonza MycoAlert detection kit) and human cell line identity was authenticated by ATCC. A549 and Vero cells were grown in DMEM, supplemented with 10% FBS, 100 U/ml penicillin, and 100 µg/ml streptomycin. Calu3 and Caco2 cells were grown in MEM, supplemented with 10% FBS, 100 U/ml penicillin, 100 µg/ml streptomycin, 1% MEM NEAA, and 1 mM sodium pyruvate.

## Plasmids and lentivirus infection

Overexpression constructs pEGPN-GFP, pEGPN-ACE2, and pEGPN-TMPRSS2 were constructed by Gibson cloning using NEBuilder master mix (New England Biolabs, E2621) with overlapping PCR generated inserts for promoter EF1α, the gene of interest, promoter PGK, and neomycin/resistance gene. Lentiviral vectors were co-transduced with MD2G and PAX2 in 293T cells (5 million cells/10 cm plate) with 25 µl of XtremeGene9 (Millipore Sigma, #6365787001) and supernatant was harvested at 48 and 72 hr post transfection. Target cells were transduced with the addition of 4 µg/ml polybrene (Santa Cruz, sc-134220). Infected cells were selected and maintained in 500 µg/ml G418 (Life Technologies, #10131027). lentiCas9-Blast was a gift from Feng Zhang (*Sanjana et al., 2014*) (Addgene plasmid #52962). ipUSEPR was a gift from Francisco Sanchez-Rivera and Scott Lowe. sgRNAs were selected from the Brunello CRISPR database (*Doench et al., 2016*). Four guides per gene were tested in Caco2 cells and the most efficient two sgRNAs/gene were used in subsequent experiments (*Figure 4—figure supplement 1*). Knockout efficiency was determined by next-generation amplicon sequencing (Genewiz).

## Pseudovirus

Recombinant VSV-based and HIV-1-based SARS-CoV-2 pseudovirus was generated as described previously (*Schmidt et al., 2020*). To generate rVSVΔG/SARS-CoV-2 pseudovirus, 293T cells (12 million cells/15 cm plate) were transfected with 12.5 µg pSARS-CoV-2$_{Δ19}$, and 24 hr post transfection, were infected with VSV-G-complemented rVSVΔG virus at an MOI of 1. Supernatant was collected 16 hr post infection, centrifuged at 350 $g$ × 10 min, filtered through a 0.45 µm filter, and concentrated using Lenti-X-Concentrator (Takara Bio). Prior to infection of target cells, the viral stock was incubated with anti-VSV-G antibody (3 µg/ml) for 1 hr at 37°C to neutralize contaminating rVSVΔG/NG/NanoLuc/VSV-G particles.

To generate HIV-1$_{NL}$/SARS-CoV-2 pseudovirus, 293T cells (12 million cells/15 cm plate) were co-transfected with 15.75 µg CCNanoLuc/GFP, 15.75 µg HIV-1$_{NL}$ GagPol, and 5.625 µg CMV-SARS-CoV-2-S, using 50 µl per 15 cm plate X-tremeGENE 9 (Sigma-Aldrich, 8724121001). Media was changed at 24 hr post transfection, and supernatant was collected at 48 and 72 hr. Centrifuged and filtered pseudovirus was concentrated with Lenti-X-Concentrator or with 40% (w/v) PEG-8000, 1.2 M NaCl, pH 7.2.

## Incucyte

Cells were imaged and analyzed using an Incucyte ZOOM (Essen BioScience). Four fields of view per well were averaged and 3–6 wells/condition were assayed in each experiment. Confluence was calculated from brightfield images, GFP/NeonGreen object confluence was calculated from green fluorescent images taken with 400 ms exposure time, and GFP+ fractional area is the ratio of these variables.

## Luciferase assay

Following pseudovirus infection, cells were washed twice with PBS, which was subsequently aspirated. Lysis buffer (Promega, E1531) was added (50 µl/well) and incubated rotating for 15 min at room temperature. NanoGlo Substrate (Promega, N1130) was diluted 1:50 in assay buffer and 25 µl/well was added and incubated for an additional 15 min. Samples were transferred to a white, opaque-bottom 96-well plate and luminescence was read using a BMG Labtech FLUOstar Omega plate reader.

## Clotting assays

Pooled normal human plasma was obtained from Pacific Hemostasis (VWR #95059–698). Factor X and prothrombin-deficient plasma were obtained from Haematologic Technologies (#FX-ID and #FII-ID). Russell's viper venom test was performed with 10 µg/ml snake venom from *Vipera russelli* (RVV, Sigma-Aldrich #V2501) diluted in Tris buffer (20 mM Tris-HCl, 150 mM NaCl, 14 mM CaCl$_2$, pH 7.5). Pre-warmed plasma was mixed with pre-warmed dilute venom (100 µl each) and monitored for visible clotting at 37°C. Prothrombin time was determined by mixing 100 µl pre-warmed plasma with 200 µl pre-warmed thromboplastin (VWR #95059–802) and monitoring for visible clotting at 37°C.

## hPSC lung organoids

The hPSC-derived lung organoids were differentiated and cultured as described previously (*Han et al., 2021*). Briefly, hPSCs were differentiated into endoderm in serum-free differentiation (SFD) medium (DMEM/F12 (3:1) (Life Technologies) supplemented with 1 × N2 (Life Technologies), 1 × B27 (Life Technologies), 50 µg/ml ascorbic acid, 2 mM Glutamax (Gibco), 0.4 µM monothioglycerol and 0.05% BSA) in a 5% O$_2$ incubator; followed by 10 µM Y-27632, 0.5 ng/ml human BMP4 (R&D Systems), 2.5 ng/ml human bFGF and 100 ng/ml human activin A (R&D Systems) for 3 days; and subsequently single cells were plated on fibronectin-coated plates. Differentiation to anterior foregut endoderm was performed with SFD with 1.5 µM dorsomorphin dihydrochloride (R&D Systems) and 10 µM SB431542 (R&D Systems) for 3 days, and then 10 µM SB431542 and 1 µM IWP2 (R&D Systems) treatment for 3 days. Differentiation to early stage lung progenitor cells was accomplished by treatment with 3 µM CHIR99021 (CHIR, Stem-RD), 10 ng/ml human FGF10, 10 ng/ml human KGF, 10 ng/ml human BMP4, and 50–60 nM all-trans retinoic acid (ATRA) in a 5% CO$_2$/air incubator. Differentiation to late-stage lung progenitor cells, cells were treated with SFD containing 3 µM CHIR99021, 10 ng/ml human FGF10, 10 ng/ml human FGF7, 10 ng/ml human BMP4 and 50 nM ATRA on fibronectin-coated plates, and subsequently maintained for 1 week in SFD medium containing 3 µM CHIR99021, 10 ng/ml human FGF10 and 10 ng/ml human KGF, in a 5% CO$_2$/air incubator. Mature lung organoids were generated by growing late-stage lung progenitor cells in 90% Matrigel in SFD medium containing 3 µM CHIR99021, 10 ng/ml human FGF10, 10 ng/ml human KGF, 50 nM dexamethasone, 0.1 mM 8-bromo-cAMP (Sigma-Aldrich), and 0.1 mM IBMX (3,7-dihydro-1-methyl-3-(2-methylpropyl)-1*H*-purine-2,6-dione; Sigma-Aldrich) for ~20 days.

## SARS-CoV-2 virus infection

SARS-CoV-2 was maintained and infections were performed as described previously (*Tang et al., 2021b*). SARS-CoV-2, isolate USA-WA1/2020 (NR-52281), was deposited by the Center for Disease Control and Prevention and obtained through BEI Resources, NIAID, NIH. SARS-CoV-2 was propagated in Vero E6 cells in DMEM supplemented with 2% FBS, 4.5 g/l D-glucose, 4 mM L-glutamine, 10 mM non-essential amino acids, 1 mM sodium pyruvate, and 10 mM HEPES. MOI of SARS-CoV-2 was determined by plaque assay in Vero E6 cells in Minimum Essential Media supplemented with 2% FBS, 4 mM L-glutamine, 0.2% BSA, 10 mM HEPES and 0.12% NaHCO$_3$, and 0.7% agar.

All work involving live SARS-CoV-2 was performed in the CDC/USDA-approved BSL-3 facility of the Global Health and Emerging Pathogens Institute at the Icahn School of Medicine at Mount Sinai in accordance with institutional biosafety requirements.

At 24 or 48 hpi, RNA was extracted with TRIzol and Direct-zol RNA Miniprep Plus kit (Zymo Research). SARS-CoV-2-N and SARS-CoV-2-E transcripts were quantified by two-step qRT-PCR using LunaScript RT SuperMix Kit (E3010L) for c-DNA synthesis and Luna Universal qPCR Master Mix (NEB #M3003) for RT-qPCR. qRT-PCRs were performed on CFX384 Touch Real-Time PCR Detection System (Bio-Rad). Primers specific for the *N* gene (SARS-CoV-2-N-F: TAATCAGACAAGGAACTGATTA, SARS-CoV-2-N-R: CGAAGGTGTGACTTCCATG), for the *E* gene (SARS-CoV-2-E-F: ACAGGTACGTTAATAG TTAATAGCGT, SARS-CoV-2-E-R: ATATTGCAGCAGTACGCACACA), as well as internal control human 18S (Forward: GGCCCTGTAATTGGAATGAGTC, Reverse: CCAAGATCCAACTACGAGCTT) were used. The delta-delta-cycle threshold ($\Delta\Delta$CT) was determined relative to 18S and vehicle-treated samples.

## Acknowledgements

The authors would like to thank Pilar Mendoza (Rockefeller University), John Blenis (WCMC), Elena Piskounova (WCMC), Tim McGraw (WCMC), Marco Straus (Cornell University), Tomer Yaron (WCMC), and all members of the Cantley Lab for insightful discussion and helpful comments. We thank Paul Bieniasz (Rockefeller University) and Theodora Hatziioannou (Rockefeller University) for providing reagents and helping to establish the pseudovirus platform in our laboratory. We thank Danielle Bulaon (Weill Cornell) for provisioning inhibitors, Benjamin tenOever (Mount Sinai) for providing Vero cells, and Francisco Sanchez-Rivera and Scott Lowe (MSKCC) for the ipUSEPR plasmid. This work was funded in part by the National Institute of Health research grants R01AI35270 (to GW) and R35 CA197588 (to LCC) and the Pershing Square Foundation (LCC).

## Additional information

### Competing interests

Robert E Schwartz: is on the scientific advisory board for Miromatrix Inc and is a consultant and speaker for Alnylam Inc. Lewis Cantley: is a founder and member of the SAB of Agios Pharmaceuticals and a founder and former member of the SAB of Ravenna Pharmaceuticals (previously Petra Pharmaceuticals). These companies are developing novel therapies for cancer. Holds equity in Agios. Lewis Cantley's laboratory also received some financial support from Ravenna Pharmaceuticals. The other authors declare that no competing interests exist.

### Funding

| Funder | Grant reference number | Author |
|---|---|---|
| National Institutes of Health | R01AI35270 | Gary R Whittaker |
| National Cancer Institute | R35CA197588 | Lewis Cantley Lewis Cantley |
| Pershing Square Foundation | | Lewis Cantley Lewis Cantley |

The funders had no role in study design, data collection and interpretation, or the decision to submit the work for publication.

### Author contributions

Edward R Kastenhuber, Conceptualization, Investigation, Methodology, Writing – original draft, Writing – review and editing; Marisa Mercadante, Vasuretha Chandar, Investigation; Benjamin Nilsson-Payant, Javier A Jaimes, Investigation, Methodology; Jared L Johnson, Methodology, Writing – review and editing; Frauke Muecksch, Methodology, Resources, Writing – review and editing; Yiska Weisblum, Methodology, Resources; Yaron Bram, Investigation, Writing – review and editing; Gary R Whittaker, Funding acquisition, Supervision, Writing – review and editing; Benjamin R tenOever, Funding acquisition, Supervision; Robert E Schwartz, Funding acquisition, Investigation, Supervision, Writing – review and editing; Lewis Cantley, Conceptualization, Funding acquisition, Supervision, Writing – original draft, Writing – review and editing

## Author ORCIDs

Edward R Kastenhuber https://orcid.org/0000-0002-1872-212X
Javier A Jaimes https://orcid.org/0000-0001-6706-092X
Frauke Muecksch https://orcid.org/0000-0002-0132-5101
Yiska Weisblum https://orcid.org/0000-0002-9249-1745
Robert E Schwartz https://orcid.org/0000-0002-5417-5995
Lewis Cantley https://orcid.org/0000-0002-1298-7653

## Decision letter and Author response

Decision letter https://doi.org/10.7554/eLife.77444.sa1
Author response https://doi.org/10.7554/eLife.77444.sa2

---

## Additional files

### Supplementary files

• Transparent reporting form

• Supplementary file 1. Graphical abstract: Positive feedback in SARS-CoV-2 infection and coagulopathy. In this study, we investigated the role of coagulation factors in SARS-CoV-2 infection. Hyperactivated coagulation is a feature of COVID-19 pathology. Coagulation factors, including factor Xa and thrombin, can cleave SARS-CoV-2 spike. This activity can exacerbate infection by enhancing viral entry. Lastly, we show that a subset of protease inhibitors with anticoagulant properties, such as nafamostat, also have the potential to block host-mediated spike activation by multiple human proteases.

### Data availability

All new plasmids will be made available through Addgene.

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
