## [Editor Report]

This study examines the potential role host proteases involved in coagulation may play in proteolytic processing of the SARS CoV-2 virus spike protein, which is required for viral entry. Serine protease inhibitors such as nafamostat and camostat may limit viral entry into host cells and also be useful to treat coagulopathy in patients with SARS CoV-2 infection, particularly if treatment is initiated early.

---

## [Decision Letter]

[Editors' note: this paper was reviewed by Review Commons.]

---

## [Author Response]

Reviewer #1 (Evidence, reproducibility and clarity (Required)):This paper shows that cleavage of SARS-CoV-2 spike protein is necessary for infection and that this is mediated by host proteases. Specifically, they show that activated coagulation factors are effective at cleaving spike protein.Major commentsThe authors clearly show that coagulation factors, especially FXa and thrombin, are capable of cleaving spike protein, as is TMPRSS2. They also show that the protease inhibitor nafamostat is effective at inhibiting spike protein cleavage.

We appreciate the Reviewer’s positive comments about our manuscript and we are grateful that the Reviewer concurs that our conclusions are clearly shown by the experimental data.

The evidence for a role for coagulation proteins in vivo is weak. Despite the use of anti-coagulants as standard of care the evidence for their benefit is weak and even studies using higher doses have not been very promising. This suggests that TMPRSS2 is the important enzyme and as long as it is uninhibited spike protein cleavage will still occur.

The Reviewer rightly points out the inherent challenge of reconciling clinical observation, which suffer from limited control over important variables and lack of molecular detail, with controlled perturbation experiments, which require compromises in approximating the physiological setting. We made our best effort to be precise in the text to not overextend our claims beyond what can be supported by the data and, where appropriate, to cite the copious but circumstantial clinical evidence that supports the physiological relevance of dysregulated coagulation in COVID-19.

A safe and effective drug regimen to address COVID-19 coagulopathy has not been optimized and is urgently needed. As the reviewer points out, multiple clinical trials failed to detect a reduction in organ support-free days following therapeutic dose over prophylactic dose heparin (A. Investigators et al., 2021; R.-C. Investigators et al., 2021). Timing is a critical factor in the interpretation of the potential for direct antiviral effects of anti-coagulation strategies. In the aforementioned studies, treatment occurred in-hospital, typically during the inflammatory phase of disease, after the peak of viral titer has already passed (Griffin et al., 2021). Thus, any antiviral effect of DOACs, for instance, are difficult to evaluate in this context. We observed that apixaban reverses the effect of exogenous FXa, but its overall effect was partial, presumably since it spares the activity of both furin and TMPRSS2 (Supplementary Figure 9). For this reason, we would predict that heparin or commonly used DOACs might have a modest antiviral effect if applied soon after disease onset, but that other drugs with a dual mechanism towards coagulation proteases and TMPRSS2/TTSPs, would be a far superior strategy.

Our results demonstrate that factor Xa can activate SARS-CoV-2 spike in cells with or without TMPRSS2 expression (Figure 3G) and that dual inhibition of factor Xa and TMPRSS2 with nafamostat can have a remarkable effect on spike-mediated entry (Figure 4 G-H, Supplementary Figure 9) without impacting furin (Supplementary Figure 8A). Our interpretation is that TMPRSS2 is one important host protease in the process of SARS-CoV-2 infection, but is unlikely to be the only important host protease. It remains challenging to fully reproduce the complex biological process of coagulation in cell culture and therefore care must be taken when extrapolating in vitro results to the human condition. Experiments with psuedoparticles and enzymatic assays using peptide substrates allow for precisely controlled conditions and provide mechanistic insight for the involvement of coagulation factors in SARS-CoV-2 spike-mediated entry.

During the course of revision, we added an additional experiment that we believe has strong physiological relevance. We generated lung organoids by stepwise differentiation of human pluripotent stem cells, which recapitulates more of the form and function of human lung tissues than lung-derived cell lines grown in standard culture conditions. We used this material to conduct an infection experiment with WT SARS-CoV-2 in lung organoids to achieve greater physiological relevance. For details, please refer to Methods (lines 664-710). We describe this experiment in the main text as follows (lines 261-277):

“Factor Xa and thrombin increase SARS-CoV-2 infection in lung organoids

To explore the effect of coagulation factors in a more physiologically relevant setting, we exposed human pluripotent stem cell-derived lung organoids (hPSC-LOs) to SARS-CoV-2 infection under BSL-3 conditions with or without addition of exogenous protease. […] This effect was accentuated at 48 hours post infection with respect to 24 hours post infection, and with lower initiating MOI. Factor Xa and thrombin increase SARS-CoV-2 infection in the context of multicycle viral replication in human lung organoids.”

Furthermore, there is significant evidence to show that coagulopathy associated with COVID-19 and other severe viral infections is in fact platelet driven (Cox D. Targeting SARS-CoV-2-Platelet Interactions in COVID-19 and Vaccine-Related Thrombosis. Frontiers in Pharmacology. 2021;12). An interesting fact about nafamostat and the related protease inhibitor pentamidine is that they are both GPIIb/IIIa antagonists (Cox D, Aoki T, Seki J, Motoyama Y and Yoshida K. Pentamidine is a specific, non-peptide, GPIIb/IIIa antagonist. Thromb Haemost. 1996;75:503-9). SARS-CoV-2 spike protein has an RGD sequence possibly allowing it to bind to platelet GPIIb/IIIa. Thus, nafamostat has three potential benefits in COVID-19: inhibition of spike protein cleavage, inhibition of SARS-CoV-2 binding to platelet GPIIb/IIIa and inhibition of platelet aggregation and thus thrombus formation.

We appreciate the Reviewer’s comments. We will incorporate the points raised and the references mentioned above into the discussion (lines 328-331 and 356-358), concerning the model of a platelet-driven COVID-19 coagulopathy and how this relates to an additional potential mechanism of nafamostat, respectively.

Line 328-331:

“Furthermore, one model posits that COVID-19 coagulopathy is platelet-driven and an Arg-Gly-Asp (RGD) motif on SARS-CoV-2 spike directly interacts with GPIIb/GPIIIa integrins on the surface of platelets (Cox, 2021), consistent with in silico predictions of integrin binding (Meszaros et al., 2021).”

Line 356-358:

“It is also plausible that nafamostat, like related protease inhibitor pentamidine, could also interfere with platelet GPIIb/IIIa, platelet aggregation and thrombus formation (Cox, 2021; Cox et al., 1996).”

Reviewer #1 (Significance (Required)):This is an interesting insight into the role of proteases in SARS-CoV-2 and may be a potential therapeutic strategy in COVID-19.I am a pharmacologist so my expertise is not in biochemistry but have a major interest in viral-mediated thrombosis.

We thank the Reviewer for their time, consideration, and expertise. We hope that readers will also be interested in the potential role of coagulation proteases in SARS-CoV-2 and how this reframes the approach to protease inhibitors currently under clinical investigation.

Reviewer #3 (Evidence, reproducibility and clarity (Required)):The authors present a study exploring the potential that host proteases involved in coagulation play a role in proteolytic processing of the spike protein on the SARS CoV-2 virus required for viral entry into target cells. They propose broad based serine protease inhibitors such as nafamostat and camostat could be useful in limiting viral entry into host cells as well as reducing coagulopathy related problems in patients with SARS CoV-2 infection particularly if treatment is initiated early in the disease course. The authors use FRET based enzymatic assays in purified systems to demonstrate that peptides mimicking the S1/S2 and S2'cleavage sites of the SARS CoV-2 spike protein can be cleaved by coagulation proteases FXa and thrombin at rates comparable to that of TMPRSS2. TMPRSS2 has been established in the relevant literature to be an important host protease involved in the proteolytic activation of several coronaviruses. In addition, the authors present data from cell based pseudoviral infection assays to demonstrate the ability of coagulation proteases to process the particles for cell entry as well as the effectiveness of inhibitors such as nafamostat and camostat in preventing infection by SARS CoV-2 pseudovirons.The data presented provides convincing support for the conclusions reached by the authors. Suitable control experiments using modified FRET peptides with key residues altered to make them poorer substrates for FXa and thrombin provided additional weight to support the conclusion that coagulation proteases could play a role in processing the spike protein for viral entry.

We appreciate the reviewer’s positive comments about our manuscript and we are grateful that the reviewer concurs that our conclusions are convincingly supported by the experimental data and that appropriate controls were included.

Prior studies seem to be referenced appropriately. The text and figures are clear and accurate.

We thank the reviewer for remarking on the appropriate contextualization of other studies in the field and the accuracy and clarity of the text and figures.

The authors may consider including phospholipids in the assay where they measure activity of FXa toward its "known" substrate THRBR271 to reduce the potential for overstating the preference for the S1/S2 peptide over natural FXa substrates.

We thank the Reviewer for raising this point. We agree with that the results of factor Xa enzymatic assays are dependent on assay-specific conditions including pH and buffer composition. We aimed to clarify our statement on FXa substrate selectivity to reflect the reviewer’s suggestion (lines 163-165, 178-180):

Line 163-165:

“factor Xa showed ~9-fold greater maximum initial reaction velocity (Vmax) in cleaving the spike S1/S2 peptide compared to cleaving a peptide corresponding to its known substrate, THRB^R271^”

Line 178-180:

“In summary, the coagulation serine proteases factor Xa and thrombin exhibit proteolytic activity against SARS-CoV-2 peptide substrates.”

Furthermore, we conducted the factor Xa enzymatic assays with the addition of phosphatidylcholine/phosphatidylserine (PC/PS) to determine the effect on factor Xa substrate preference. The results were added to Supplementary Figure 5 (related to figure 2). We found that the addition of phospholipid vesicles did not change factor Xa activity or substrate preference in this assay (Figure S5E). To ensure the quality of our phospholipid vesicle preparation, we added 0-100 µM PC/PS to a dilute Russell’s Viper Venom time clotting assay, where PC/PS drives a significant, concentration-dependent acceleration of clotting of normal pooled human plasma (Figure S5E). The dRVVT test was selected over more common coagulation assays because it relies on direct activation of factor X to Xa and does not rely on reagents that include exogenous phospholipids.

Our interpretation is that the use of purified, RVV-activated factor Xa and peptide substrates in the enzymatic assay renders the reaction independent of phospholipid membranes. On the surface of platelets, one might expect that providing a membrane scaffold, thereby spatially restricting the tenase complex that activates factor Xa, could potentially increase the velocity of the cleavage activation of factor X, by increasing the likelihood of collision of fX and tenase (Basavaraj and Krishnaswamy, 2020; Husten, Esmon, and Johnson, 1987). However, our assay relies on purified factor Xa, activated by Russell’s viper venom (Tans and Rosing, 2001), alleviating the requirement for upstream activation. The use of peptide substrates in this assay has the consequence that downstream cleavage requires that enzyme and substrate meet in 3-dimensional space, rather than along the surface of a 2-dimensional membrane.

Lines 590-600 (Methods):

“Phospholipid vesicles

Phosphatidylcholine (1,2-dioleoyl-sn-glycero-3-phosphocholine, Avanti Polar Lipids #850375C) and phosphatidylserine (1,2-dioleoyl-sn-glycero-3-phospho-L-serine, Avanti Polar Lipids #840035C) were mixed in a 3:1 w/w ratio in chloroform solvent in a screw top vial and the chloroform solvent was evaporated under a nitrogen stream. Unilamellar vesicles were isolated by extrusion using 0.1 micron pore filters and diluted in buffer AB2 (50mM Tris-HCl, 150mM NaCl, pH 8). The addition of phospholipid vesicles did not change factor Xa activity or substrate preference in this assay (Figure S5E). To ensure the quality of our phospholipid vesicle preparation, we added 0-100 µM PC/PS to a dilute Russell’s Viper Venom time clotting assay, where PC/PS drives a significant, concentration-dependent acceleration of clotting of normal pooled human plasma (Figure S5E).”

Reviewer #3 (Significance (Required)):Given the rapidly evolving nature of the SARS CoV-2 pandemic the work presented in the manuscript should be relevant to both the fields of virology and coagulation. The potential for treatments aimed at inhibiting both TMPRSS2 and coagulation associated proteases should be of interest to both clinicians and basic science researchers involved in the treatment and research of SARS CoV-2.

We appreciate that the Reviewer found the manuscript to be relevant and of broad interest to the fields of virology and coagulation. We also hope that the work is of interest to both clinicians and basic scientists.

References:

Basavaraj, M. G., and Krishnaswamy, S. (2020). Exosite binding drives substrate affinity for the activation of coagulation factor X by the intrinsic Xase complex. *J Biol Chem, 295*(45), 15198-15207. doi:10.1074/jbc.RA120.015325

Cox, D. (2021). Targeting SARS-CoV-2-Platelet Interactions in COVID-19 and Vaccine-Related Thrombosis. *Front Pharmacol, 12*, 708665. doi:10.3389/fphar.2021.708665

Cox, D., Aoki, T., Seki, J., Motoyama, Y., and Yoshida, K. (1996). Pentamidine is a specific, non-peptide, GPIIb/IIIa antagonist. *Thromb Haemost, 75*(3), 503-509. Retrieved from https://www.ncbi.nlm.nih.gov/pubmed/8701416

Griffin, D. O., Brennan-Rieder, D., Ngo, B., Kory, P., Confalonieri, M., Shapiro, L.,... Marik, P. (2021). The Importance of Understanding the Stages of COVID-19 in Treatment and Trials. *AIDS Rev*. doi:10.24875/AIDSRev.200001261

Husten, E. J., Esmon, C. T., and Johnson, A. E. (1987). The active site of blood coagulation factor Xa. Its distance from the phospholipid surface and its conformational sensitivity to components of the prothrombinase complex. *J Biol Chem, 262*(27), 12953-12961. Retrieved from https://www.ncbi.nlm.nih.gov/pubmed/3477541

Investigators, A., Investigators, A. C.-a., Investigators, R.-C., Lawler, P. R., Goligher, E. C., Berger, J. S.,... Zarychanski, R. (2021). Therapeutic Anticoagulation with Heparin in Noncritically Ill Patients with Covid-19. *N Engl J Med, 385*(9), 790-802. doi:10.1056/NEJMoa2105911

Investigators, R.-C., Investigators, A. C.-a., Investigators, A., Goligher, E. C., Bradbury, C. A., McVerry, B. J.,... Zarychanski, R. (2021). Therapeutic Anticoagulation with Heparin in Critically Ill Patients with Covid-19. *N Engl J Med, 385*(9), 777-789. doi:10.1056/NEJMoa2103417

Meszaros, B., Samano-Sanchez, H., Alvarado-Valverde, J., Calyseva, J., Martinez-Perez, E., Alves, R.,... Gibson, T. J. (2021). Short linear motif candidates in the cell entry system used by SARS-CoV-2 and their potential therapeutic implications. *Sci Signal, 14*(665). doi:10.1126/scisignal.abd0334

Tans, G., and Rosing, J. (2001). Snake venom activators of factor X: an overview. *Haemostasis, 31*(3-6), 225-233. doi:10.1159/000048067